# Inferring the regional distribution of Visceral Leishmaniasis incidence from data at different spatial scales
Emily S. Nightingale [1,2] ✉, Swaminathan Subramanian [3], Ashley R. Schwarzer [4], Lloyd A. C. Chapman [5], Purushothaman Jambulingam[3], Mary M. Cameron[6], Oliver J. Brady [1,2], Graham F. Medley [2,7] & Tim C. D. Lucas[8]

## Abstract

**Background** As cases of visceral leishmaniasis (VL) in India dwindle, there is motivation to monitor elimination progress on a finer geographic scale than sub-district (block). Low-incidence projections across geographically- and demographically- heterogeneous communities are difficult to act upon, and equitable elimination cannot be achieved if local pockets of incidence are overlooked. However, maintaining consistent surveillance at this scale is resource-intensive and not sustainable in the long-term.

**Methods** We analysed VL incidence across 45,000 villages in Bihar state, exploring spatial autocorrelation and associations with local environmental conditions in order to assess the feasibility of inference at this scale. We evaluated a statistical disaggregation approach to infer finer spatial variation from routinely-collected, block-level data, validating against observed village-level incidence.

**Results** This disaggregation approach does not estimate village-level incidence more accurately than a baseline assumption of block-homogeneity. Spatial auto-correlation is evident on a block-level but weak between neighbouring villages within the same block, possibly suggesting that longer-range transmission (e.g., due to population movement) may be an important contributor to village-level heterogeneity.

**Conclusions** Increasing the range of reactive interventions to neighbouring villages may not improve their efficacy in suppressing transmission, but maintaining surveillance and diagnostic capacity in areas distant from recently observed cases - particularly along routes of population movement from endemic regions - could reduce reintroduction risk in currently unaffected villages. The reactive, spatially-targeted approach to VL surveillance limits interpretability of data observed at the village level, and hence the feasibility of routinely drawing and validating inference at this scale.

## Plain Language Summary

Near elimination, it is important to understand how the remaining cases of disease are distributed on a local level. However, surveillance data are more easily collated according to larger administrative units. We investigated whether village-level patterns of visceral leishmaniasis (VL) incidence could be inferred from administrative-level data using a statistical modelling approach. We found strong similarity in incidence between neighbouring administrative units but not between neighbouring villages, and model predictions did not correspond well to observed village-level case data. This could suggest that longer-range transmission contributes more to the village-level pattern of incidence than short in this near-elimination context, which should be considered in intervention planning. However, increased surveillance effort in assumed high-risk villages makes interpretation of data at this level challenging.

A key issue raised in previous work forecasting Visceral Leishmaniasis (VL) incidence at the block level is the appropriateness of this geographic scale of inference for drawing actionable conclusions. Spatial correlation in block-level incidence was observed[1] and it was demonstrated that exploiting these correlations had value for improving short-term temporal predictions. The block, however, is too large of a scale for targeting low levels of transmission and incidence; predictions at a higher spatial resolution are needed. It has been demonstrated previously that the choice of spatial scale/units of analysis can have an unintended influence on conclusions (known as the Modifiable Areal Unit Problem, or MAUP). In brief, this is the problem that

[1]Department of Infectious Disease Epidemiology and Dynamics, London School of Hygiene and Tropical Medicine, London, UK. [2]Centre for Mathematical Modelling of Infectious Diseases, London School of Hygiene and Tropical Medicine, London, UK. [3]ICMR—Vector Control Research Centre, Puducherry, India. [4]London School of Hygiene and Tropical Medicine, London, UK. [5]School of Mathematical Sciences, Lancaster University, Lancaster, UK. [6]Department of Disease Control, London School of Hygiene and Tropical Medicine, London, UK. [7]Department of Global Health and Development, London School of Hygiene and Tropical Medicine, London, UK. [8]Department of Population Health Sciences, University of Leicester, Leicester, UK. ✉e-mail: emily.nightingale@lshtm.ac.uk

a model of village-level data can yield different conclusions to a model of block-level data. Careful consideration is therefore required as to what partition is appropriate given how the data have been collected[2].

On the other hand, an elimination setting inevitably demands a reduction in the intensity of surveillance in order to be sustainable in the long term, and monitoring incidence at a fine scale is incredibly resource-intensive. The population of Bihar state is spread across approximately 45,000 villages - from densely populated wards of the capital city to remote, rural hamlets. In particular across the region south of the Ganges river, the vast majority of villages are not affected by VL. However, an important minority of villages suffer persistent outbreaks.

Case counts of VL are currently monitored at the village level to inform vector control and case detection activities[3], yet villages are treated almost entirely independently; recent observation of any case in a village dictates subsequent years' interventions in that village alone, but not neighbouring villages. This approach to the deployment of active case detection appears to capture a majority of future cases[4] yet inevitably cannot account for sporadic cases in previously unaffected villages. It could be more efficient to apply interventions within a certain range of a persistently-affected village, rather than waiting for a case to be detected in each individual village in order to trigger a response. The present work therefore also aims to evaluate the evidence for correlation between incidence observed in neighbouring villages, to ascertain whether the efficacy and efficiency of this intervention could be improved by broadening its spatial range.

Clustering of cases within villages has been previously demonstrated[5–7], while correlation between villages has primarily been explored with respect to climatic and environmental conditions suitable to the sand fly vector[5,8–10]. Transmission of VL occurs when adult female sand flies seek human blood to mature their eggs, therefore conditions for sand fly breeding will influence the exposure of the human population. In particular, the type of vegetation, temperature, moisture and living conditions of the human population have been suggested as potentially related to transmission risk[11,12]. Such studies have however been limited in spatial scale to one or two example districts, usually chosen due to high disease burden (or low, to serve as a control).

The analysis presented here aims to draw inference from routinely reported VL diagnoses at the block level, combining this with remotely-sensed covariate data and exploiting a disaggregation approach[13] to infer the potential distribution of those cases at a more local level. Also sometimes referred to as "downscaling", methods for inferring fine scale variation from spatially aggregated data have progressed substantially in recent years, alongside computational developments in the field of spatial statistics more broadly[14–22].

It is, however, rarely possible to validate disaggregation approaches against data actually observed at the finer scale. Python et al.[18] were able to exploit two sources of data available at the district level to validate their disaggregation of province-level COVID-19 incidence, but neither had complete country-wide coverage. Previous validation of this particular implementation had only been conducted by simulation[17]. For the case described here, acquisition of GPS coordinates for VL-affected villages in Bihar has made it possible to attribute observed cases to a precise location in space and to infer the locations of unaffected villages through linkage with village boundary polygons. These data provide a unique opportunity to evaluate whether disaggregation can accurately replicate the distribution of a block's case count across its constituent villages, for the entire state.

It is increasingly inefficient to implement uniform interventions across broad geographic units as incidence continues to decline and transmission may be limited to a few small pockets of the population. Identifying and enumerating each unique village in the state of Bihar is a complex and resource-intensive exercise, yet it is now routine to collect a GPS location for each newly-diagnosed case. Knowing the relative locations villages allows us to acknowledge similarities between nearby villages - with respect to both population and transmission risk - to inform the use of targeted interventions, but is more challenging analytically than current practice.

Here we evaluate an approach which does not depend on the collection and maintenance of surveillance data at the village level, with an aim to assess the added value of this information for our understanding of the spatial distribution of observed disease burden, and potentially of underlying transmission. We find that the approach does not estimate village-level incidence more accurately than a baseline assumption of block-homogeneity. Spatial auto-correlation is evident on a block-level but weak between neighbouring villages within the same block, possibly suggesting that longer-range transmission (e.g., due to population movement) may be an important contributor to village-level heterogeneity.

## Materials and methods
### Data
Counts of VL cases diagnosed in 2018 per village in Bihar were compiled and shared by CARE India, along with coordinates for the centroid of each village affected. These data were first linked to corresponding village polygons by overlaying the affected village point locations. Where multiple points fell within the same polygon, case counts were aggregated. Polygons in which no points fell were defined as unaffected and attributed with a case count of zero.

Populations were estimated by first extracting and summing 100 m pixel values from the WorldPop UN-adjusted population count raster for 2018[23] for the set of village polygons. The counts for constituent villages were then aggregated, alongside case counts and polygon geometries, to yield a block-level (administrative level 3) analysis dataset with which to fit the disaggregation model. Pixel-level predictions from the disaggregation model could then be aggregated according to the same village polygons and validated against the original village level counts (Fig. 1).

**Pixel-level covariates.** Potentially predictive spatial covariates were identified by reviewing previous analyses of risk factors for VL incidence on a local population/community level, and characteristics of suitable sandfly habitats[24–26]. These broadly fell under the categories of elevation, rurality and moisture in the environment (proximity to water, vegetation and temperature). Publicly-available data sources were then identified to capture these characteristics at a pixel level with highest resolution.

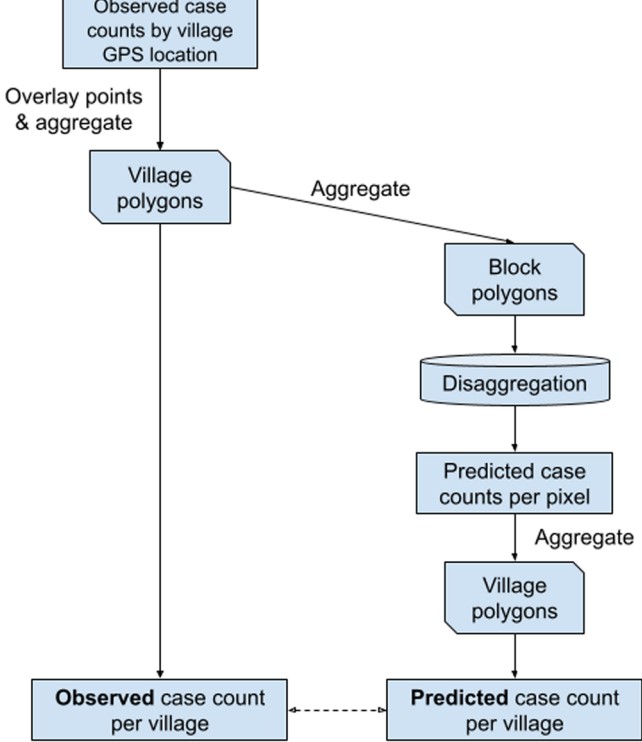

**Fig. 1 |** Data processing steps to perform and validate block-level disaggregation, based on the available geotagged village case counts.

https://doi.org/10.1038/s43856-024-00659-9 **Article**

Raster data for elevation (metres above sea level) and distance (in metres, as of 2015) to inland water bodies were obtained from WorldPop at a resolution of 100m for the region of Bihar state[27]. Estimated travel time (in minutes) to the nearest urban centre as of 2015 was obtained from the Malaria Atlas Project (MAP) at a resolution of 1 km[28], with an urban centre defined as a "contiguous area with 1500 or more inhabitants per square kilometre, or a majority of built-up land cover coincident with a population centre of at least 50,000 inhabitants". Land surface temperature (LST) and normalised difference vegetation index (NDVI) at a 1 km resolution were extracted for the same region from MODIS/Terra satellite data, accessed via the AppEEARS platform[29–31]. The latter two were initially extracted on a monthly scale for the period of the case data (2018–2019) and subsequently aggregated to an annual mean and standard deviation. All covariate rasters were resampled to the lowest resolution (1km) and scaled to account for the inconsistent units of measurement. No uncertainty in these inferred covariate values was incorporated into the analysis.

## Data cleaning

The raw village incidence data included sixty villages out of 2186 affected during 2018 which were missing GPS coordinates and hence could not be directly linked to a village polygon, 40 of which fell within only three blocks (Barauli, Bhorey and Kuchaikote in Gopalganj district). As far as possible, these villages were manually matched to polygons according to district, block, gram panchayat (a local unit of usually multiple villages) and village names; five villages (7 cases) were unidentifiable and hence excluded.

Two villages had GPS coordinates which placed them substantially outside the state boundary; data errors were identified in the latitude variable and corrected. A further 8 villages had coordinates which fell marginally outside the boundary; only two of these had reported cases which were attributed to the nearest village polygon, with a tolerance of 500 m (Supplementary Fig. S2).

When aggregating the population count raster to these polygons, 83 village shapes (0.2%) were calculated to have zero population. None of these were attributed with any reported cases and were therefore ignored in comparison of incidence rates.

## Descriptive analysis

Preliminary analyses assessed the evidence for (global) spatial auto-correlation in incidence between neighbouring villages for the year 2018, by calculation of Moran's I statistic. This was calculated across all villages in the state and separately across villages within each block independently, to explore whether the strength of correlation between nearby villages could differ between blocks. The strength of evidence for auto-correlation was interpreted by comparison of the observed statistic value to the distribution of values calculated from 999 permutations of the data under the assumption of spatial independence[32].

## Disaggregation model structure

Disaggregation regression combines observed block-level case counts with these finer-scale population and covariate data to predict the potential within-block distribution of incidence. For this analysis, we evaluate the Bayesian implementation described in ref. 13 and published in the R package *disaggregation*.

The model is specified as a regression on the pixel level, with covariate values predicting case counts per pixel. However, the case counts per pixel are not known, and the model parameters are instead optimised relative to the sum of pixel counts across areas (in this case blocks).

For incidence rate $r$ in pixel $j$ in block $i$ with location $s_{ij}$,

$$\log(r_{ij}) = \beta_0 + \beta X_{ij} + GRF(s_{ij}) + u_i \qquad (0.1)$$

where $X_{ij}$ are covariate values for pixel $j$ in block $i$, GRF is a Gaussian random field with Matérn covariance function defined across pixels $s_{ij}$, and $u_i$ is a block-level uncorrelated random effect.

The case count in block $i$ is then obtained by aggregating $r_{ij}$ via a weighted raster $a_{ij}$ (i.e., the population raster) across all pixels $j \in 1, \ldots, N_i$ in block $i$,

$$cases_i = \sum_{j=1}^{N_i} a_{ij} r_{ij} \qquad (0.2)$$

This is finally linked to the observed case count in block $i$ through a Poisson likelihood, i.e., $y_i \sim Pois(cases_i)$.

A Poisson likelihood is a natural choice for a count outcome, and is mathematically convenient for the aggregation step within the disaggregation model structure. If we assume that the number of cases observed in each pixel follows a Poisson distribution, then it follows that the sum across pixels also follows a Poisson distribution. It is often the case that greater variation is observed in the outcome of interest than can be accommodated with a Poisson distribution (overdispersion). One option to address this would be to use a negative binomial likelihood, with a fixed dispersion parameter to accommodate the additional variation which is estimated across all blocks. In this example, however, heterogeneity across blocks in terms of geographic area, population size, and other characteristics would mean that assuming the same scale of extra-Poisson variation for all blocks is likely too simplistic. The fitted IID component provides the flexibility to absorb residual extra-Poisson variation in block-level counts that is not explained by the given covariates.

Two variants of the disaggregation model were fitted—both including and excluding the spatial random field. As the field offers greater flexibility in representing a spatial pattern, it was expected that this component would be more influential in the model if the selected covariates were not strongly informative. Substantial differences in the estimated fixed effects when the field is included or excluded could indicate that there are important features of the observed spatial pattern that are only represented by the random field, with the selected covariates appearing influential only as proxies in the absence of other information.

Pixel level estimates of incidence from the disaggregation model were weighted by the 1km population raster and aggregated according to the percentage coverage of each pixel overlapping each village shape, using the R package *exactextractr*[33]. The resulting estimated case counts over each village polygon were then rescaled by the estimated village populations for comparison with observed incidence rates. For polygons in which the estimated village population was exactly zero, the incidence rate was defined as zero.

## Model fitting

This model structure requires a non-standard estimation procedure as the predictions to be obtained are at a different scale to the response data. In other words, the number of rows of covariate data differs from the number of rows of response data. The disaggregation package defines the joint likelihood function and prior model in C++, which is then passed to Template Model Builder (TMB). TMB then uses a series of packages for automatic differentiation, linear algebra and computation of sparse matrices, to implement the automatic Laplace approximation[34] to obtain an approximation to the Bayesian posterior.

This approach draws on and adapts the approach described in ref. 35 for modelling spatial processes using the integrated nested Laplace approximation (INLA). Notably, under this approach the hyperparameters are handled similarly to an empirical Bayesian framework and therefore uncertainty in these parameters are not fully propagated to the posterior. The Laplace approximation is based on an assumption that the posterior distribution is multivariate Gaussian, therefore estimates are presented with 95% Gaussian credible intervals calculated from the estimated standard errors. Experiments in ref. 13 and ref. 18 have shown that the posteriors of the model's parameters and hyper parameters are well approximated by

Gaussian distributions, using smaller datasets than that which is considered here. The priors for each parameter and hyperparameter to be estimated are defined in the Supplementary Methods.

## Validation

The strength of the disaggregation approach for predicting village level incidence was interpreted relative to two alternative benchmarks. First, the observed block-level incidence rate was defined as a baseline prediction for all villages within the block. This reflects the accuracy of assuming village-level incidence based on crude block-level surveillance, with no further information on heterogeneity between villages other than population size.

Secondly, a random forest model[36,37] was fit directly to the village level data (using the R package *randomforest*) to serve as a 'gold-standard' for predicting village-level incidence in the hypothetical scenario that village-level data could be made routinely available to guide decision-making. Random forest is a non-parametric approach commonly used to map spatially-varying phenomena due to its ability to accommodate complex non-linearities and interactions among given predictors (for example, it is the methodological basis for WorldPop's global population estimates[15]). Our intention was to employ the flexibility of this approach to maximise the information gleaned about the pattern of village-level incidence from the given data.

A random forest model is formed of an ensemble of decision trees, within each of which the training data are partitioned according to splits defined on the given predictors in order to minimise the variation in the outcome for observations within each partition. The predicted value for a new observation is defined as the average outcome of all training observations in the partition within which it falls. Spatial structure will be incorporated by including the latitude and longitude of each village centroid as predictors. To increase robustness against over-fitting, a random subset of predictors are considered when determining each split. The sensitivity of the fit to the size of this subset will be assessed by comparison of three alternatives, considering two, three, and six predictors out of the total of ten.

The baseline and disaggregation models were both evaluated against the village-level data which were not used for fitting. The random forest model, however, was fit directly to the village-level data, therefore, a cross-validated measure of predictive power was required. This measure can be obtained using "out-of-bag" (OOB) observations; each decision tree within the model is trained with a random subset of observations, therefore predictions can be defined for each observation by averaging the predictions of every tree from which the point was excluded. In this case, each observation was excluded (and hence predicted out-of-bag) an average of 74 times across the 200 fitted trees.

The correlation (Spearman's rank correlation), root mean squared error (RMSE) and mean absolute error (MAE) were used to compare

between observed village-level incidence and the baseline, OOB random forest and disaggregation-based predictions. Approximate confidence intervals for the rank correlations were calculated as suggested by ref. 38.

**Sensitivity to population estimates.** For most villages affected by VL between 2013 and 2018, CARE India has estimated population sizes based on their own enumeration during routine visits. The WorldPop raster data yield village populations of broadly similar magnitude to these more accurate, locally informed estimates, but with substantial noise (Supplementary Fig. S1). The robustness of the model validation and comparison to this estimated denominator was therefore investigated by repeating the comparisons using the CARE estimates to calculate incidence, across villages for which an estimate is available.

## Ethics and permissions

Ethical approval was obtained from the London School of Hygiene and Tropical Medicine ethics committee for this specific study (ref:27487) which falls within the broader objectives of the SPEAK India research consortium https://speakindia.org.in/(ref: 14674). Permissions were granted by the National Centre for Vector Borne Disease Control in India (NCVBDC) for analysis of the KA-MIS surveillance data to address SPEAK India's research objectives.

## Statistics and reproducibility

All analyses were conducted using R version 4.2.1 (2022-06-23). Disaggregation models were fit over 1000 iterations and the random forest model fit with 200 trees. Full details of model specifications and computation options used can be found in the published code repository[39].

## Results
### Descriptive

The state of Bihar is comprised of 534 blocks and 44,794 villages in total. The primary analysis data consist of 3609 new cases of VL diagnosed throughout 2018, across 1900 villages in 332 blocks. Based on estimated village population counts, block level incidence ranged from zero to just under 6 cases per 10,000 residents (Fig. 2a). On average, villages cover an area of one to two square kilometres, while blocks are on a scale of several hundred. The vast majority of incidence was observed across a cluster of blocks in the north-west of the state (Fig. 2b), historically a persistent focal area for VL. The south of the state, partitioned by the Ganges river, observed little to no incidence across the year.

Assuming homogeneity of incidence within blocks implies substantially different expected village level case counts than were observed (Fig. 3). In particular, observed incidence is more sparse and clustered, with a greater number of villages observing more than two cases than would have

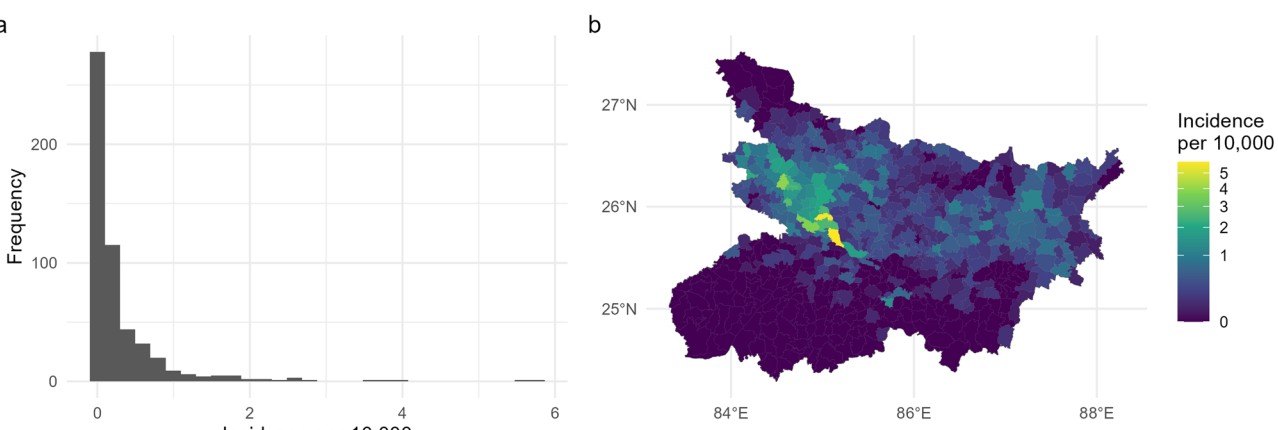

**Fig. 2 | Block level incidence of VL diagnoses per 10,000 population for 2018** (*N* = 332 blocks with at least one diagnosed case). **a** Overall distribution of block-level incidence rates, and **b** the same rates mapped to block shapefiles. The vast

majority of blocks saw zero or very low levels of incidence, in particular across the south of the state. A cluster of blocks in the west and (to a lesser extent) the east experienced moderate to high levels of incidence.

**Fig. 3 | Observed village incidence for 2018 compared to that which would be expected assuming uniformity of incidence across each block (block incidence rate multiplied by village populations size, rounded to the nearest whole case).** Note that the log scaling on the y-axis exaggerates the under-estimation of higher case counts compared to the over-estimation of villages with zero or one cases.

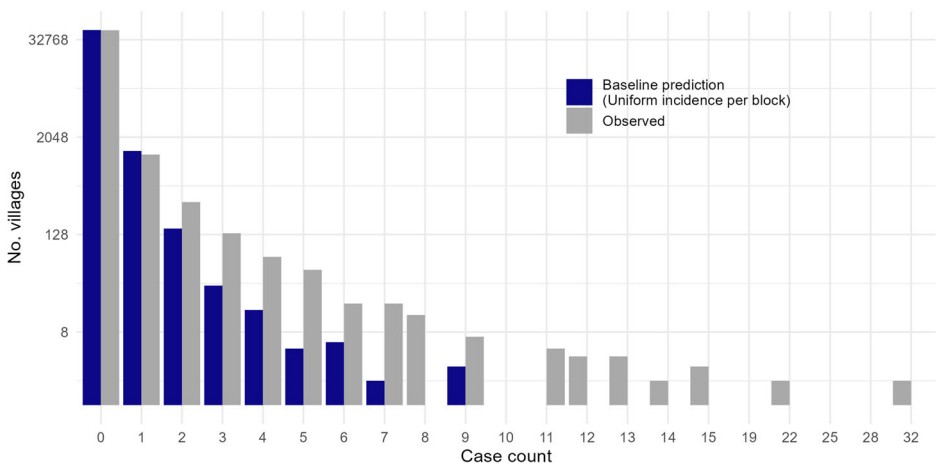

**Table 1 | Summary of estimated population size and covariates included in the disaggregation model, averaged across village polygons and further stratified by village VL status for 2018 (affected/unaffected)**

| Variable | | Overall (*N* = 44,794) | Affected (*N* = 1900) | Unaffected (*N* = 42,894) |
|---|---|---|---|---|
| Population size | | 1398.8 [595.1, 3077.1] | 3934.8 [2064.4, 8096.2] | 1339.1 [571.3, 2916.7] |
| Elevation (metres above sea level) | | 62.92 [51.4, 82.84] | 57.41 [50.37, 64.59] | 63.48 [51.48, 84.22] |
| Distance to nearest water body (kilometres) | | 0.86 [0.49, 1.61] | 0.58 [0.37, 0.9] | 0.88 [0.5, 1.65] |
| Travel time to nearest urban centre (minutes) | | 11.75 [5.63, 19.59] | 9.73 [4.94, 15.88] | 11.87 [5.66, 19.78] |
| Land surface temperature (degrees celsius) | Mean | 30.27 [29.25, 31.2] | 29.73 [28.84, 30.39] | 30.3 [29.28, 31.23] |
| | SD | 5.45 [4.68, 6.34] | 4.85 [4.41, 5.27] | 5.5 [4.7, 6.38] |
| Normalised difference vegetation index (range 0–1) | Mean | 0.48 [0.45, 0.52] | 0.48 [0.44, 0.51] | 0.48 [0.45, 0.52] |
| | SD | 0.16 [0.14, 0.18] | 0.16 [0.14, 0.18] | 0.16 [0.14, 0.18] |

Summary values are median [IQR]. Note that all covariate values (excluding the offset, population size) are standardised for model fitting.

been be expected from block level incidence rates. This interpretation is supported through calculation of Moran's I statistic, which demonstrates substantial evidence of spatial auto-correlation in observed incidence on the scale of both villages and blocks (Supplementary Fig. S3). When evaluated by each block individually, the strength of correlation between constituent villages did not appear to correlate with overall block incidence (Supplementary Fig. S4) and in fact very few blocks gave an indication of correlation between their constituent villages.

**Covariates.** When averaged across village polygons, no clear differences were apparent between affected and unaffected villages with respect to the included covariates, from a univariate perspective (Table 1). See Supplementary Fig. S5 for the raw spatial distribution of all included covariates.

**Disaggregation model fit**
The smooth spatial field contributes substantially to the overall model fit, attenuating much of the effect of the covariates and rendering the corresponding coefficients as insignificant with respect to the 95% credible intervals (Fig. 4a). A fit based only on covariates and the block-level IID effect suggests that greater VL incidence at the village level is associated with closer proximity to water, lower annual variation in temperature, and lower annual average and greater variation in the vegetation index. The only association for which significance persists in the full model is that with annual variation in NDVI, with greater variation being associated with greater village incidence. Figure 4b, c illustrates the predicted per-pixel case count from the full disaggregation model and the fitted spatial field.

**Model validation and comparison**
Upon aggregating these predictions and comparing to observed village level incidence, neither version of the disaggregation model improved on a

baseline prediction applying the block-level incidence rate (Fig. 5). Out-of-bag predictions from the village-level random forest model attained a lower RMSE than the baseline, but were the weakest with respect to the other metrics considered. This pattern persists when blocks with zero observed cases are excluded from the calculations.

The difference in ranking between RMSE and MAE suggests that the disaggregation models make some larger errors than the random forest model (which are penalised more strongly by RMSE), even if overall the predictions are closer to the observed values. As expected, the errors and correlation increase and decrease, respectively, with the exclusion of villages within zero-count blocks. As these form a substantial proportion of the total villages, accurately predicting zero cases here has a strong influence on the summary measure.

Assessing predictions individually, the observed and predicted magnitude of incidence in non-zero villages showed some linear correspondence but with substantial noise (Fig. 6). Disaggregation visually appeared to yield somewhat greater discrimination between affected and unaffected villages than the baseline. Supplementary comparison based on CARE's population estimates demonstrates even weaker correlation between observed and disaggregation-predicted values (Supplementary Fig. S6).

The random forest fit was only estimated to have explained approximately 6.6% of the variation in observed incidence, which decreased with the number of variables tried at each split (to a minimum of 2% when all ten variables were used). The distribution of out-of-bag predictions more closely replicated the observed than the disaggregation-based predictions (Fig. 7), but still under-predicted overall. Overall, all the methods considered here fail to account for the strong clustering of cases and instead redistribute cases too evenly across villages in a block, resulting in under-estimation of the possible number of cases within a village.

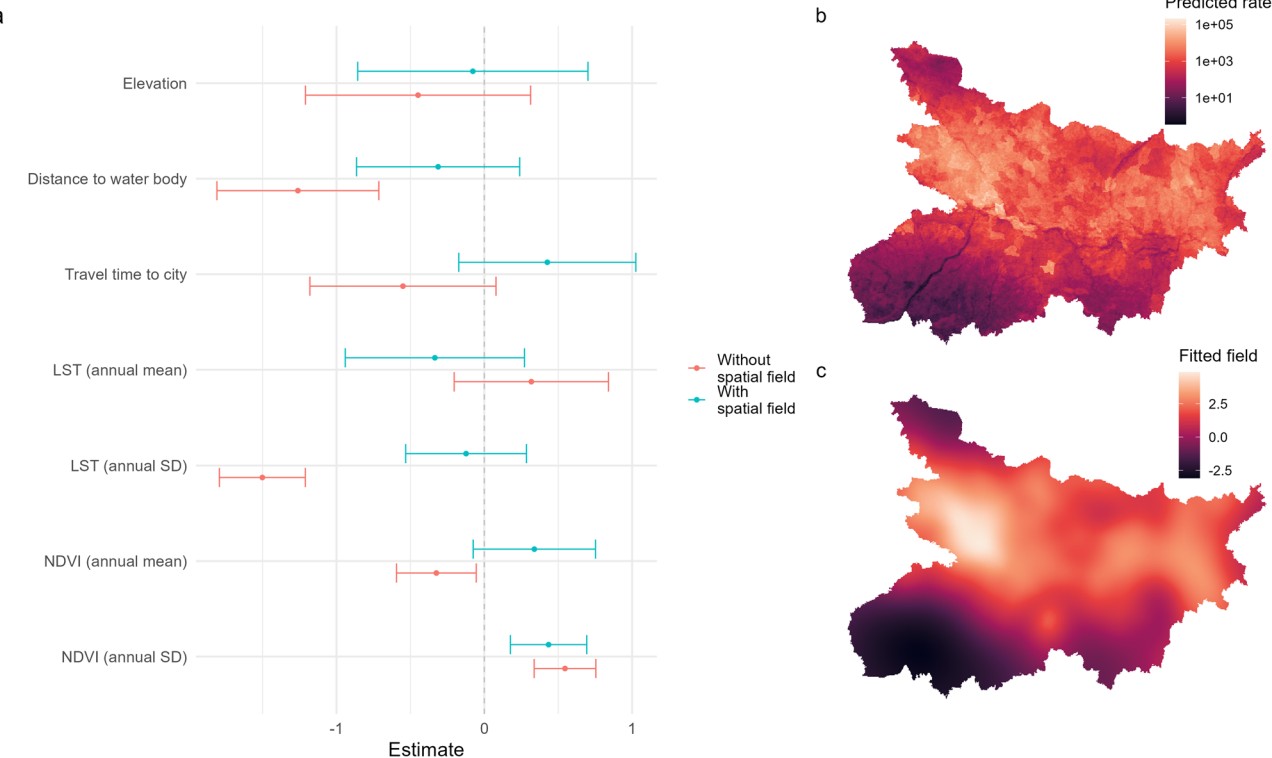

**Fig. 4 | Summary of disaggregation model fits. a** Estimated covariate coefficients (log-scale) from disaggregation model fits (fit to total case counts from 534 blocks, over 12 months), with and without the smooth spatial field. Point estimates are presented with approximate 95% credible intervals. **b** Predicted pixel-level incidence and **c** fitted spatial field from the full disaggregation model.

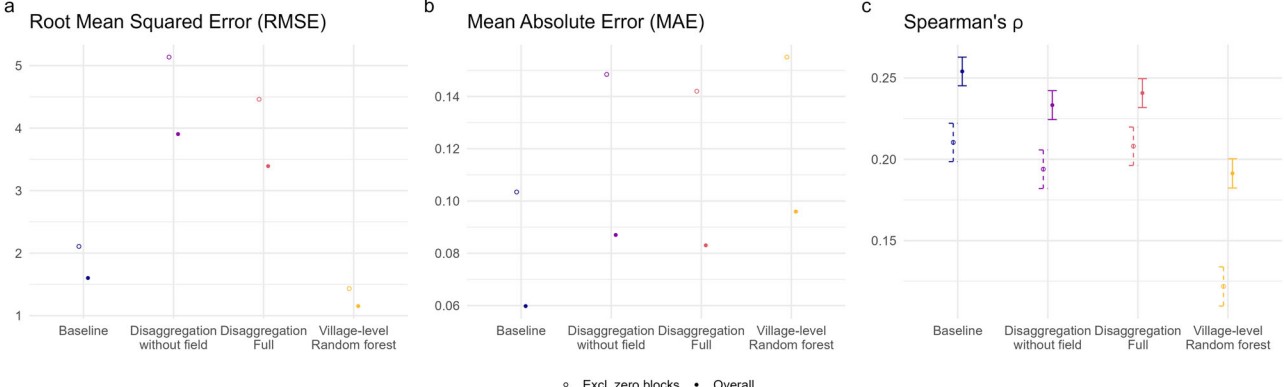

**Fig. 5 | Model comparison of predictive accuracy with respect to root mean squared error (RMSE). a** Root mean squared error, **b** Mean absolute error, and **c** Spearman's ρ, summarised over predictions for 44,794 villages. Overall measures are presented alongside alternatives excluding villages within blocks with zero total cases. Approximate 95% confidence intervals are illustrated for Spearman's ρ.

## Discussion

The analysis presented here is, as far as we are aware, the first state-wide analysis of village-level VL incidence in Bihar. The work of CARE India's field teams to enumerate the villages of Bihar and geo-locate those affected by VL has created an opportunity to explore spatial patterns in incidence on a much finer scale than has previously been feasible. Disaggregation regression provides an opportunity to interrogate fine scale variation from the type of administrative level surveillance data which is routinely available in many endemic / elimination settings. In this example, the approach was not found to be effective for estimating village level burden of VL from block level surveillance data. Moreover, even when fitting a model directly to village-level data and allowing for more complex, non-linear relationships with the local environmental conditions, it was still not possible to accurately predict incidence at withheld villages.

Evidence of heterogeneity is observed within blocks at the village level; however, a simple assumption of uniformity within blocks crudely captures the broader spatial patterns across the state and therefore still provides somewhat reasonable predictions of village level incidence overall. Preliminary investigation of spatial auto-correlation at the two scales supports the idea that patterns of correlation are evident on the broader, block-level but not necessarily between neighbouring villages. As has been demonstrated previously[11,40], we observed that cases were clustered within villages, with three or more cases observed in substantially more villages than would be expected from uniform within-block incidence. This did not, however, appear to be informative of incidence in the surrounding villages. Our results are instead consistent with relatively rare village-level clusters occurring apparently at random on a local scale, with some longer-range spatial pattern potentially driven by the environment and/or population factors.

**Fig. 6 | Comparison of predicted to observed village incidence rates (N = 44,794 villages), with respect to magnitude and presence/absence.** Scatter plots only include affected villages (N = 1900), with non-zero observed and predicted incidence. Grey lines illustrate a simple linear trend (and 95% CI) of observed against predicted. The x-axes in both columns are limited between 1e-5 and 750 per 1000.

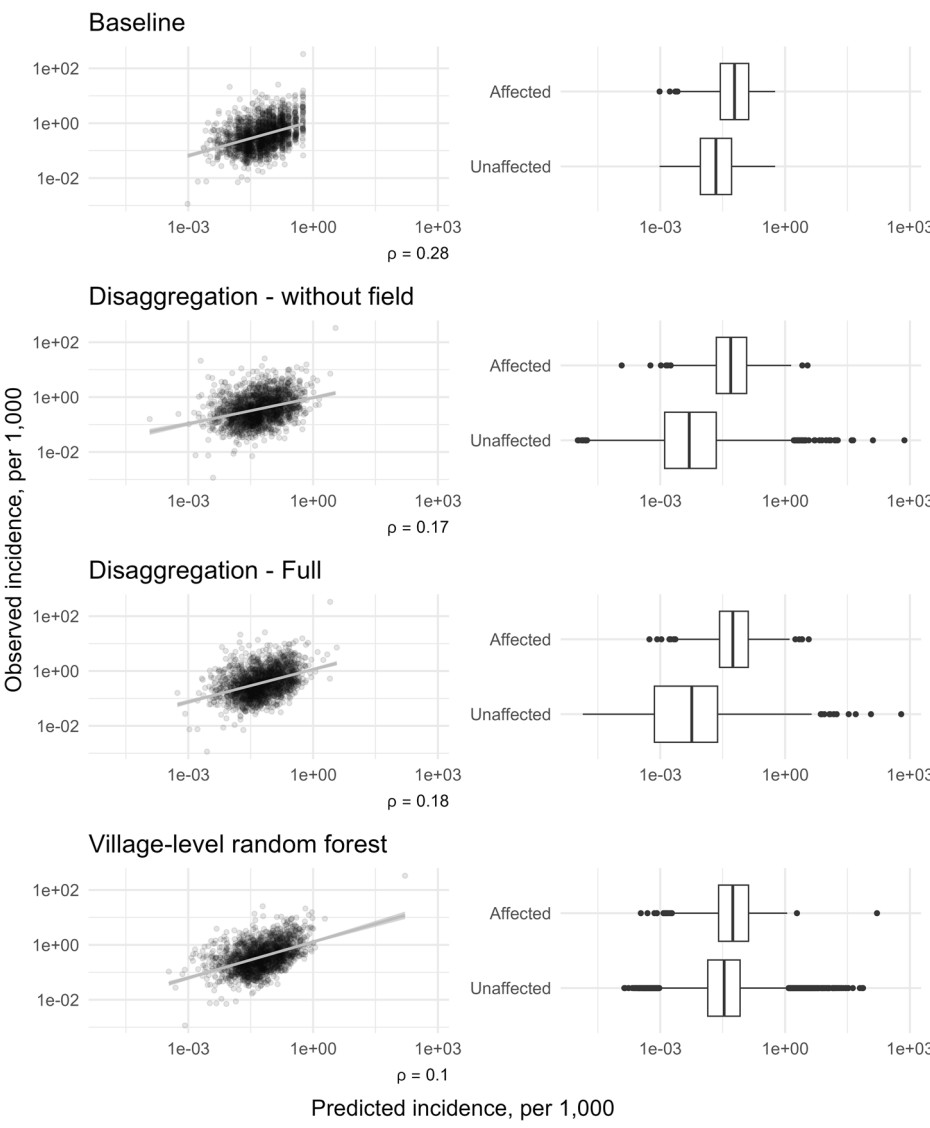

Previous work estimating the spatial range of sand fly movement and of human-to-human transmission supports the same conclusion that a long range of direct transmission is unlikely[7,40,41]. Bihar has a highly mobile population[42], which has been linked to increased VL risk[43,44] and other health concerns for the worker and their accompanying family[45,46]. It may be the case that VL outbreaks are more often triggered by human migration and translocation from outbreaks, as opposed to infected sand flies. Consequently, longer distance movement of people may be a critical mechanism in the persistence of transmission at this stage of elimination. As such movement is incredibly difficult to quantify and anticipate, this would give rise to apparently random occurrence of village outbreaks.

There are a number of potential explanations for the poor performance of disaggregation against the simpler model. Firstly, the strength of the approach depends on associations with spatial covariates from which to infer that local variation. Despite the biological link between VL transmission and the environment via the sand fly vector, the environmental characteristics considered here were not found to have clear relationships with observed VL incidence at the village level. This could be due to a lack of variation in these characteristics on such a small scale; microclimatic conditions may operate on this level which are not represented through the satellite data. Previous geostatistical and ecological analyses of environmental risk factors have demonstrated some evidence of association across a range of variables, but within a much more limited set of locations, and in some cases only indirectly with respect to sand fly abundance rather than VL

incidence[8,12,24]. In this analysis, only annual variation in the vegetation index had a robust association with incidence. Such an association could be linked to differences in agricultural practices between higher and lower incidence regions; however, likely correlation between covariates means that individual effects should not be over-interpreted.

Socio-demographic factors will also play a role in facilitating transmission - either through increased exposure or decreased access to care - but are not usually feasible to measure or estimate on a fine and continuous spatial scale. For example, sleeping and defecating outdoors increases exposure and is more common in less affluent, rural areas where VL burden is high[6,43,47]. Such mechanisms could have been captured by travel time to the nearest urban centre, yet this was not found to be informative in either model. A more relevant measure may be travel time to a health facility which offers VL diagnosis and/or treatment, since not all public health facilities in the state are equipped to offer this. For vulnerable populations living in poverty, there will be a large financial barrier associated with this distance that could delay intervention and extend opportunity for onward transmission. It has also been suggested that cases of VL-HIV co-infection and post-kala-azar dermal leishmaniasis (PKDL) likely make an increasingly important contribution to the persistence of transmission[7,48]. It is therefore important that the spatial distribution of these conditions is also investigated.

There are examples in which area level data are combined with data collected at specific point locations (for example from prevalence surveys)

**Fig. 7 | Overall distribution of model predictions across all villages. a** Distribution of observed versus predicted case counts from each model. Models capture the number of zero and low incidence villages but underestimate villages with a higher case count. **b** Overall densities of predicted village incidence rates from each modelling approach compared to the observed (dashed line).

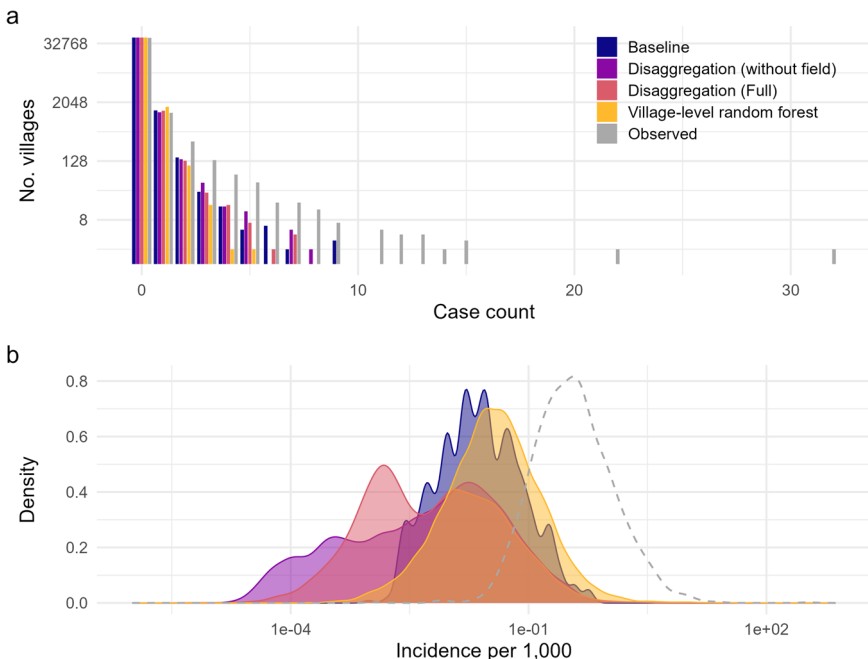

within a joint model that draws on the information of both spatial scales[16,19,21,49]. Wilson and Wakefield[49] found deterioration of accuracy when fitting only to areal census data versus point and areal, which worsened when cases were split across larger areas. Incorporating some village-level data into the disaggregation via a joint model, even if limited to a few focal locations, may improve the accuracy of prediction on this scale.

The vast majority of villages reported zero cases during this time period. Although the spatial random field will absorb this non-Poisson variation, the imbalance will mean that a wide range of environmental conditions will be represented among zero-case villages alone, making any differences with non-zero villages difficult to identify. The disaggregation framework also assumes linear covariate relationships, whereas the true underlying dynamics may be highly non-linear. To explore this possibility we also employed a random forest approach which does not depend on any such assumption, yet this did not yield an improvement on prediction.

Inferring the appropriate functional form of covariates within a spatially-indexed model is complex, since spatial patterns in covariates which drive the outcome may be easily absorbed by spatially-correlated random effects[50]. This was evident here from the change in model coefficients when a spatial field was included. There may be scope for developing the current implementation of disaggregation regression to employ restricted spatial regression as suggested in ref. 50, fitting the spatial random effects only within the residual space after adjustment for the specified fixed effects. As the goal for this analysis was prediction rather than inference of covariate associations, this was not investigated further. Lucas et al.[16] demonstrated the use of machine learning techniques to first identify relevant non-linear relationships with covariates from point-prevalence data to then feed into a disaggregation model, and found that this improved accuracy relative to a baseline using only the raw covariates.

Uncertainty in village population size influences the interpretation of village-level incidence. Surveillance field teams have estimated population size for a subset of villages during routine visits, which do not closely align with what is inferred from WorldPop's 2015 global estimates within the same village boundaries (median and IQR of 3160 [1720–5800] and 1400 [600–3080], respectively). Supplementary analyses based on locally-estimated population size resulted in weaker correlation between predicted and observed incidence, although this may be in part due to the smaller number of villages included.

Given that VL is a slow-progressing disease, there is a temporal dimension to the distribution of disease burden which cannot be interpreted by only looking at a single year of data. Infection within one village may result in transmission to surrounding villages which is not observed as clinical disease until the following year. Village incidence may therefore be highly predictive of future incidence in the surrounding area, however, we are here investigating a scenario in which data resolved spatially to the village level are not routinely available. We aimed to explore in the first instance whether the block level incidence could be disaggregated into villages for a single year, considering that if this were successful the approach could potentially be used to anticipate burden in future years. Weaknesses identified when predicting only spatially suggested to us that extending this over time would not be fruitful.

A crucial limitation of only having reported cases of disease to use as a basis for inference is that, if the processes of observation and reporting are inconsistent between villages, this would result in a poor, non-representative validation set. Village-level targeting of active case detection means that cases may be more likely to be observed in and around historically affected villages, which may induce or exaggerate patterns of spatial auto-correlation. We would not expect such patterns - driven by the observation process and not by underlying transmission - to be explained by the environmental covariates that were considered here. This targeting mechanism adds a further temporal dimension that is very challenging to untangle from transmission and disease progression, as previous incidence in a village triggers increased surveillance effort in future.

The possibility of inferring fine scale variation in disease burden from large-scale routine data through disaggregation regression would be incredibly valuable to policy makers, in particular in resource-constrained elimination settings. This analysis, however, highlights practical limitations that commonly arise with surveillance in such settings. At this stage of near-elimination of VL in Bihar, reported diagnoses of VL appear largely stochastic. It is possible that relationships between the environment and transmission that naturally arise from the underlying biological mechanisms have been broken down by intensified control efforts, patterns becoming increasingly fragmented as incidence has fallen to very low levels. Cases continue to arise in within-village clusters, yet this does not appear to be informative of the detection of cases in neighbouring villages.

We conclude that local level VL surveillance most likely is necessary for effective targeted interventions, but that the value of this information is largely in the ability to rapidly respond and detect secondary cases village by village, rather than in the ability to then anticipate incidence in the surrounding area. A fully geographically-targeted approach does not seem

feasible given the stochastic nature of incidence that we have observed. Maintaining surveillance and VL diagnostic capacity even in areas apparently distant from recent transmission is therefore critical, with a particular focus on high-risk routes of population movement from endemic to non-endemic regions.

## Reporting summary

Further information on research design is available in the Nature Portfolio Reporting Summary linked to this article.

## Data availability

Tables of results underlying the model comparison (aggregated error metrics) and associated figures (observed versus predicted distribution of counts across all villages) are published in the following repository https://github.com/esnightingale/vl-disaggregation[39]. The VL incidence data (block- and village-level) are the property of the National Centre for Vector-Borne Diseases Control (NCVBDC, Ministry of Health and Family Welfare, Government of India) and requests to access these data should be made directly to the Centre.

## Code availability

The code used to perform this analysis is also published in the above repository https://github.com/esnightingale/vl-disaggregation[39].

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

## Acknowledgements
The authors offer their profound gratitude to the field teams at CARE India who work tirelessly to collect and organise the case details and spatial data without which this work would not have been possible. Also, we would like to thank Sridhar Srikantiah, Joy Bindroo and Khushbu Priyamvada for their support in managing and sharing their understanding of the data and the underlying detection processes. Finally, our thanks go to the National Centre for Vector Borne Diseases Control for their permission to use these data.

## Author contributions
E.S.N. conceptualised the study, performed the analyses, wrote the manuscript and produced the remote code repository. T.C.D.L., O.J.B. and G.F.M. provided supervision regarding the methodology, study concept, presentation and interpetation of results. S.Su, P.J. and L.A.C.C. provided feedback on the study concept and manuscript. A.S. conducted preliminary work on construction and analysis of the village level data, supervised by L.A.C.C. and E.S.N.; M.M.C. supervised as PI of the SPEAK India consortium. All co-authors provided feedback on the manuscript and approved the final submitted version.

## Competing interests
The authors declare the following competing interests. This work was funded by the Bill and Melinda Gates Foundation (ESN, LCC, MMC:OPP1183986, GFM: OPP1184344). O.J.B. was funded by a UK Medical Research CouncilCareer Development Award (MR/V031112/1). T.C.D.L. was supported by the NIHR Applied Research Collaboration East Midlands (ARC EM). The views expressed are thoseof the author(s) and not necessarily those of the NIHR or the Department of Health and Social Care. S.Su and P.J. are funded by the Indian Council of Medical Research via the Vector Control Research Centre.
