## [Transparent Peer Review file · Communications Medicine]

Inferring the regional distribution of Visceral Leishmaniasis incidence from data at different spatial scales

Corresponding Author: Dr Emily Nightingale

Version 0:

Reviewer comments:

Reviewer #1

(Remarks to the Author)

Inferring the regional distribution of Visceral Leishmaniasis incidence from data at different spatial scales

The researchers point at a need for surveillance for VL in India below the block level. They explored village level VL incidence across the State of Bihar and associations with local environmental conditions, taking into account spatial auto-correlation. They found that this approach did not estimate village incidence more accurately than assuming homogenous incidence at block level. Spatial auto-correlation was evident at block level but less so between neighboring villages within blocks. There may be important transmission mechanisms acting at a longer range. Expanding interventions to neighboring villages may not increase efficacy but improving surveillance and treatment among mobile populations could be more beneficial.

Major comments:

This is a well written manuscript, though I am not a modeler so I'm not able to assess the technicalities related to mathematical modeling. I just want to raise a few questions about the practical implications for public health. Do I understand correctly that the researchers conclude that based on the models they fitted there is no evidence for clustering above village level and that the co-variables explored (NDVI, climatological variables etc.) cannot really predict in which village the next cluster is likely to occur?

The researchers recommend 'surveillance among mobile population groups at risk of (re-) introducing infection to previously unaffected villages'. Are migrants really a risk for (re)introducing VL in a village? To the best of my knowledge migrants in Bihar tend to migrate to big cities or out of country, e.g. to the Gulf states. Do the researchers agree that migrants would only pose a risk if they would move between VL endemic areas, i.e. villages in Bihar/Jharkhand? It might be stigmatizing to assume that migrants are the cause of local outbreaks and targeting migrants with interventions might not offer the best value for money.

Minor comments:

The researchers conclude that: 'local level VL surveillance most likely is necessary for effective targeted interventions, but that the value of this information is largely in the ability to rapidly respond and detect secondary cases village by village, rather than in the ability to then anticipate incidence in the surrounding area'. Did the researchers also consider the sub-village level (tola) because it appears that VL in Bihar clusters at that level?

Considering the fact that VL may pop up anywhere, should there not be a recommendation for availability of diagnostics (RDTs) in all blocks within (previously) endemic districts?

Reviewer #2

(Remarks to the Author)

Review of "Inferring the regional distribution of Visceral Leishmaniasis incidence from data at different spatial scales"

Firstly, I would like to extend my apologies for the delay in delivering this peer review. I want to emphasise that my peer review aims to support your work and assist you as authors in making your research as impactful as possible in academic and policy forums. While I have provided several recommendations, please remember that this is your paper, and you have the freedom to develop it according to your vision.

General Comment

1. The introduction and discussion sections of this paper are generally well-written and clear, with only a few sentences that may benefit from rephrasing to enhance conciseness. Additionally, the authors present nice plots and figures throughout the paper.
2. Most of the comments I provide here are based on the methodology, particularly on validating the disaggregation model with other models such as the baseline model and the RF model. I think the authors need to reconsider the spatial unit for fitting the model. For example, in using the RF model, the authors fit their model at the village level and disaggregate to the pixel level, but in the baseline and the disaggregation model, they fit the model at the block level. Obviously, the observational spatial unit for model fitting will produce very different results; hence, model validation based on these differences will introduce inherent biases. Ideally, all the models should have the same spatial unit for analysis and validation.

Introduction:

Overall, I don't have much to say regarding the introduction. I think the authors provide a comprehensive overview of the context and motivations for the study in inferring the regional distribution of Visceral Leishmaniasis (VL) incidence from data at different spatial scales.

However, here are a few comments and suggestions on the introduction.

3. Page 2, lines 48–52: the authors mention the MAUP. I think it would be ideal to explain further what the MAUP is for a non-technical audience and readers who may not be familiar with the concept.
4. I think the writing in general is clear and concise, but there are a few instances where the phrasing could be improved for clarity and flow. E.g.
Line 47: "The vast majority of these villages are not affected by VL, in particular those south of the Ganges River, while others persistently observe cases and suffer outbreaks."
Line 65: "Interventions could be applied more efficiently if sporadically-affected villages were covered within the range of a nearby persistently-affected village, rather than waiting for a response to be triggered within each independently".

Method:

5. On page 4, line 118, the authors stated their utilisation of population data from WorldPop. However, WorldPop offers various types of population data. Could the authors please specify which data product they utilised and from which year it was extracted? Clarification on this detail would enhance the comprehensiveness of the methodology and aid in reproducibility.
6. On page 4, line 126, it would be beneficial to clarify the year of the covariates utilised in the study. Additionally, the authors may want to provide justification for the selection of these covariates as input variables. How many covariates were considered, and was there a specific criterion for eliminating certain covariates? Furthermore, were there any concerns regarding multicollinearity among the covariates? Addressing these questions regarding covariate selection and potential issues would enhance the thoroughness and robustness of the study.
7. On page 5, lines 140–144, the assessment of spatial autocorrelation appears unclear. It would be helpful to specify whether the spatial autocorrelation of VL incidence was evaluated at the block level or the village level. Please clarify.
8. On page 6, lines 146–153, there are different approaches to disaggregating observed incidence. Is there any justification for selecting the method they used in the disaggregation model? If so, clarify. Another thing is to mention the name of the package and what software was used for the disaggregation model.
9. Also, the covariates extracted for the modelling are measured in different units. Are these covariates scaled before they are used as input in the modelling? Unscaled covariates may have implications for the disaggregated result. I would suggest that the covariates used in the model need to be scaled or standardised into a z-score or some other method to allow for easy comparison of these covariates.

Validation

10. On page 6, lines 170-174, the description of the baseline prediction model for validation lacks clarity. It is unclear whether the model utilises the observed incidence rate at the block level and disaggregates cases across all villages, regardless of whether a village has previously recorded a case. If this is the case, then it is necessary to reconsider the baseline model for validation. If a village has not reported any cases, the model should be constrained accordingly and not make predictions in such a village. Redistributing case counts to villages with no previous incidence may not be appropriate because we are redistributing case counts to villages with no incidence in reality. If there are no recorded cases in a given village why are you making predictions at the pixel location in such villages? So, I think this model needs to be reconsidered.
11. On pages 175–177, I think the authors need to reconsider the spatial scale in the use of the random forest model in validating their predictions. The authors claim they fitted a random forest (RF) model at the village level and made predictions at the village level. And this was used to validate the model. I

think we have an inherent bias with the spatial unit of analysis.

Ideally, if we want to compare how the RF model performs with our model, which is fitted at the block level, then the spatial unit of fitting the two models should be the same, i.e., the RF model should also use data observed at the block level and not at the village level as in the case of the RF. We want the spatial unit for fitting the model to be the same so that we are not introducing any bias in the predictions. So, the authors need to reconsider the RF spatial unit for fitting the model. Also, what package or software was used in fitting the RF model? A

random forest model works best when you have more covariates to partition the feature space.

I can't say how efficient the RF model will be with this small number of covariates. But I am guessing that could be a potential reason why the RF model metrics did not improve much relative to the other model. It would be worth considering other covariates in the RF model.

12. Overall, I suggest that the authors revisit their validation approach. To effectively assess the efficiency of the disaggregation model in comparison with the RF model and other models, it is essential for all models to be fitted at the same spatial unit or scale. For instance, fitting all models at the block level and subsequently disaggregating them to pixel levels would ensure consistency and minimise bias in the comparisons. Furthermore, it is advisable to constrain the models so that predictions or disaggregation are only performed for villages with recorded VL cases rather than those without any incidence. Additionally, employing a variety of model metrics for evaluation beyond just RMSE and correlation would provide a more comprehensive assessment of model performance. This approach would enhance the robustness and reliability of the study's findings.

Results

13. On page 9, Data Cleaning, I think the data cleaning section should be shifted to the methodology section and not the result section.

12. On Page 11, lines 266–268, I think the authors need to specify in the methodology the different models they fitted with the disaggregation model. No mention is made of the fitting of two separate disaggregation models in the method, but the authors in the result section are reporting two disaggregation methods, which is confusing. They need to provide clarity in the method section.

13. Page 11, table 3.2: I think the validation of the model should be reconsidered based on the comment I provided in the methodology section. Also, aside from the two metrics of correlation and RMSE, different metrics can also be calculated to give a broader perspective of how the model is performing.

Discussion

14. I think the author's discussion is well written overall. However, I think their discussion needs to be re-considered in line with key issues found within the results that relate to the methodology. I think if issues raised in the methodology are considered (absolutely at your discretion and that of the editor), then the discussion can be rewritten considering new results to communicate the new result that has been found.

Reviewer #3

(Remarks to the Author)

Summary:

The article evaluates the feasibility of using a statistical model to obtain accurate disease incidence rate estimates for individual villages based on aggregated incidence data at a coarser resolution (the block level). In particular, the authors study the incidence of visceral leishmaniasis in the Indian state of Bihar of 2018. The authors use a dataset of village level incidence rates which allows them to evaluate a disaggregation approach which only use the spatially aggregated block level data as input. The authors evaluate two geostatistical models that incorporate covariate information (derived from satellite imagery) and Gaussian process modeling, finding that the geostatistical models do not effectively downscale the aggregated data. Moreover, the authors show that even a random forest approach that utilizes the village-level data does not yield estimates that improve upon the baseline assumption of block-level homogeneity in incidence.

Overall impression:

The article is well-written, with a clear motivation and thoughtful discussion. The study's use of high-resolution spatial data to validate disaggregation approaches is highly valuable and relevant to researchers applying geostatistical modeling to estimate health-related rates, as such "gold-standard" datasets are rarely available. The presentation of the statistical and machine learning approaches used could be improved. In particular, additional justification of modeling choices and model validation would strengthen the article, as it is currently not obvious that the model used for disaggregation represents the best possible model. The discussion of results and limitations also needs editing to remove imprecise language, and the limitations related to the data (limited covariate data resolution, uncertainty related to the population estimates) should be emphasized.

Specific comments and recommendation:

1. Line 154: The Gaussian random field is not clearly specified here. The authors should describe what covariance function is used, and how the parameters for the covariance function are estimated/specified.
2. Line 158: The authors use a Poisson likelihood but do not justify this choice. Especially since there are many zeroes at the village level, would another likelihood or model be more appropriate? Is there any overdispersion observed? Some discussion of alternative choices (and subsequent justification of the model used) would improve the article.
3. Line 159-162: What are the parameters described by the posterior distribution here? It is true that the Laplace approximation relies on Gaussian assumptions for the posterior distribution, but it is not obvious that the posterior distribution

for the r_{ij} rate variables would be multivariate Gaussian. This point could be further clarified. The inferential framework should also be discussed further. The mention of a posterior distribution suggests a Bayesian approach, but no prior distribution is mentioned and usually priors are not included when applying TMB.

4. Line 164: The aggregation step could be further explained. What is done with pixels that are partially overlapping village polygons?

5. Line 178: Typo; remove second "is."

6. Line 191: Four alternatives are mentioned, but only three are provided.

7. Line 199: The claim that each observation is predicted 200 times is confusing. Are these observations held out of every tree? If a given observation is included in the training data for any of the trees, then it would be predicted fewer than 200 times.

8. Line 245: The authors note that villages cover an area of one to two square kilometers. If the covariate raster pixels are 1km by 1km, then perhaps the covariate data is simply not available at a high enough resolution to carry out disaggregation to the village level. Are there many villages that are smaller than a pixel? In such cases, how would the disaggregation procedure be justified? More discussion of how the resolution of the covariate data/prediction grid affects results is needed.

9. Line 252-253: "greater than two cases than expected" is unclear.

10. Figure 3.2: This figure is somewhat confusing. The total number of villages should be the same for both Baseline and Observed, but it looks like the total is much smaller for Baseline. Moreover, the choice of bins (of unequal size) makes the pattern look smoother than it may be in reality. I would suggest keeping all bins (0, 1, 2, 3, 4, 5, etc.) to improve interpretability.

11. Line 286-288: Do all methods always predict zero cases for villages in a block with zero cases? It seems like it would be best to leave such blocks out of the analysis as no downscaling would be necessary if the analyst knew with certainty that there would be no cases in the block. As such, the authors should also provide error metrics that leave out these blocks.

12. Line 372: What does it mean that the "spatial random field will absorb this non-Poisson variation?" This should be made more precise and rigorous.

13. Line 384-385: Restricted spatial regression is typically used in association studies where estimating associations between covariates and response is a priority. In this case, only prediction is required. Would switching to restricted spatial regression affect the results here?

14. Line 422: "incidence appears largely stochastic" is too vague of a claim, given that VL has a clear mechanism of spread and infection. It may be true that it is difficult to predict incidence given available data, but this claim seems to suggest that the disease simply occurs randomly.

Version 1:

Reviewer comments:

Reviewer #1

(Remarks to the Author)

I approve the revised version with just one small remark. In line 400 I would suggest to remove 'with many migrant workers' because it seems to be mobility within the State rather than migration that leads to spread of transmission.

Reviewer #2

(Remarks to the Author)

Reviewer #3

(Remarks to the Author)

Summary:

The authors have made a considerable effort to address issues raised in the first review and the exposition of the technical details and discussion of results are largely improved. In particular, the authors' efforts to explain their inferential framework and modeling choices are appreciated. Some issues remain; I provide some specific comments and suggestions below. The line numbers refer to the marked-up manuscript with changes tracked.

Specific comments and recommendations:

1. Line 55: The definition of the MAUP is confusing (and not correct—a finite dataset can only be partitioned and aggregated a finite number of ways). I recommend the authors precisely define the MAUP in concrete terms: for example, a model of village level data can yield different conclusions than a model of block level data.

2. Line 113: Typo: "Knowing the relative locations villages"

3. Line 203: The discussion of the Poisson likelihood is welcome. With regards to the iid random effect, the authors could mention that since the blocks are of different sizes, assuming the same variance for all blocks may be too simple. The authors note that "there was no evidence of residual extra-Poisson variation"—how was this investigated? Given this claim, it would be helpful to see parameter estimates for the various variance parameters discussed here.

4. Line 217: This comment is not particularly informative. It's true that the estimated covariate effects change depending on whether the GRF is included, due to spatial confounding. It's not clear what it means for your conclusions when you say the field "absorbs the spatial variation represented by the covariates." The authors could more clearly explain the expected result from including/excluding the spatial random field.

5. Line 238: typo: "Random forest is a non-parametric approach is commonly"
6. Figure 3.2: As far as I can tell, there are still significantly more total observed villages (summing over the heights of the gray bars) than villages as predicted by the baseline prediction (summing over the heights of the blue bars). Why is this the case?
7. Line 351 add space: "considered.This"
8. Figure 3.6: As with Figure 3.2, there are still significantly more total observed villages (summing over the heights of the gray bars) than villages as predicted by the various methods. Why is this? On another note, it may help to discuss why the observed density is shifted so far right compared with the prediction methods? It seems that these methods smooth incidence too much—so cases that in reality cluster in a small number of villages are redistributed too evenly across the villages in a block.
9. Response to author's rebuttal about TMB: It's true that TMB is typically used in a frequentist framework, but hyperparameter priors may be included in the model files. However, base TMB code does not integrate over multiple hyperparameter values—rather, when a prior is included, inference typically follows an empirical Bayesian framework. Uncertainty from the hyperparameters is not propagated to the final posterior (and indeed can make a Gaussian approximation seem more appropriate than is really the case).

Version 2:

Reviewer comments:

Reviewer #3

(Remarks to the Author)

I thank the authors for their thoughtful responses. I have no further comments.

21st June 2024

Communications Medicine

Dear Reviewers,

RE: “Inferring the distribution of Visceral Leishmaniasis incidence from data at different spatial scales”

Thank you very much for your feedback on our submitted manuscript and for offering us an opportunity to revise. Please find below our responses to your comments.

Kind Regards,

Emily Nightingale

Reviewer #1 (Remarks to the Author):

Inferring the regional distribution of Visceral Leishmaniasis incidence from data at different spatial scales

The researchers point at a need for surveillance for VL in India below the block level. They explored village level VL incidence across the State of Bihar and associations with local environmental conditions, taking into account spatial auto-correlation. They found that this approach did not estimate village incidence more accurately than assuming homogenous incidence at block level. Spatial auto-correlation was evident at block level but less so between neighboring villages within blocks. There may be important transmission mechanisms acting at a longer range. Expanding interventions to neighboring villages may not increase efficacy but improving surveillance and treatment among mobile populations could be more beneficial.

Major comments:

This is a well written manuscript, though I am not a modeler so I'm not able to assess the

technicalities related to mathematical modeling. I just want to raise a few questions about the practical implications for public health. Do I understand correctly that the researchers conclude that based on the models they fitted there is no evidence for clustering above village level and that the co-variables explored (NDVI, climatological variables etc.) cannot really predict in which village the next cluster is likely to occur?

This is correct.

The researchers recommend 'surveillance among mobile population groups at risk of (re-) introducing infection to previously unaffected villages'. Are migrants really a risk for (re)introducing VL in a village? To the best of my knowledge migrants in Bihar tend to migrate to big cities or out of country, e.g. to the Gulf states. Do the researchers agree that migrants would only pose a risk if they would move between VL endemic areas, i.e. villages in Bihar/Jharkhand? It might be stigmatizing to assume that migrants are the cause of local outbreaks and targeting migrants with interventions might not offer the best value for money.

We agree with the reviewer that the term "migrant" is not appropriate in this context, but also that - yes - a risk is posed by routine movement between endemic and non-endemic villages/areas of the state e.g. for work, shopping, visiting family, etc. We instead emphasise the need to maintain surveillance and diagnostic capacity even in areas apparently a substantial distance away from recent transmission.

[Line 469; Conclusions] *A fully geographically-targeted approach does not seem feasible given the stochastic nature of incidence that we have observed. Maintaining surveillance and VL diagnostic capacity even in areas apparently distant from recent transmission is therefore critical, with a particular focus on high-risk routes of population movement from endemic to non-endemic regions.*

This has also been updated in the abstract:

[Abstract] *Increasing the range of reactive interventions to neighbouring villages may therefore not improve their efficacy in suppressing transmission. However, maintaining surveillance and diagnostic capacity in areas distant from recently observed cases - in particular along routes of population movement out of endemic regions - could reduce the risk of reintroduction into previously unaffected villages.*

Minor comments:

The researchers conclude that: 'local level VL surveillance most likely is necessary for effective targeted interventions, but that the value of this information is largely in the ability to rapidly respond and detect secondary cases village by village, rather than in the ability to then anticipate incidence in the surrounding area'. Did the researchers also consider the sub-village level (tola) because it appears that VL in Bihar clusters at that level?

No, we did not consider tolas as a unit of analysis; the village was the smallest spatial scale at which this analysis was viable, due to the need for defined populations and

geographical boundaries across which to make predictions. Already there are challenges with defining and working across the ~50k villages of Bihar, and to move to tolas would increase that complexity an order of magnitude further. We acknowledge that previous work has identified clustering within/between tolas in the case of specific outbreaks, across a limited geographic area. Our aim here was to explore what analysis would be feasible across the entire state, from the perspective of monitoring state-wide elimination.

Considering the fact that VL may pop up anywhere, should there not be a recommendation for availability of diagnostics (RDTs) in all blocks within (previously) endemic districts?

Yes absolutely. Not all health facilities have capacity for VL diagnosis and treatment which could lead to less complete and prompt detection of infection. As stated in response to the point above, we have now emphasised this point in place of the attention to mobile populations (*Line 435; Conclusions*).

Reviewer #2 (Remarks to the Author):

Review of “Inferring the regional distribution of Visceral Leishmaniasis incidence from data at different spatial scales”

Firstly, I would like to extend my apologies for the delay in delivering this peer review. I want to emphasise that my peer review aims to support your work and assist you as authors in making your research as impactful as possible in academic and policy forums. While I have provided several recommendations, please remember that this is your paper, and you have the freedom to develop it according to your vision.

Thank you for taking the time to provide thoughtful feedback.

General Comment

1. The introduction and discussion sections of this paper are generally well-written and clear, with only a few sentences that may benefit from rephrasing to enhance conciseness. Additionally, the authors present nice plots and figures throughout the paper.

Thank you.

2. Most of the comments I provide here are based on the methodology, particularly on validating the disaggregation model with other models such as the baseline model and the RF model. I think the authors need to reconsider the spatial unit for fitting the model.

For example, in using the RF model, the authors fit their model at the village level and disaggregate to the pixel level, but in the baseline and the disaggregation model, they fit the

model at the block level. Obviously, the observational spatial unit for model fitting will produce very different results; hence, model validation based on these differences will introduce inherent biases.

Ideally, all the models should have the same spatial unit for analysis and validation.

To clarify, the RF model was fit directly to the village-level data and *not* disaggregated at all; i.e. fit to village level in order to predict at village level, using cross-validation to avoid over-estimation of accuracy. The baseline and disaggregation models are both fit to the block level in order to predict at village level, with the latter disaggregating to pixel level predictions which are then *re-aggregated* to village.

As stated in the title of the manuscript, the aim of this analysis is *exactly* to compare the predictions of models fit to data at different spatial scales. The purpose of this is to understand whether prediction from coarser but more readily-available block-level data can match the accuracy of prediction from finer scale data. For this comparison of predictive accuracy, predictions from all models are evaluated against the same set of village level incidence data.

Introduction:

Overall, I don't have much to say regarding the introduction. I think the authors provide a comprehensive overview of the context and motivations for the study in inferring the regional distribution of Visceral Leishmaniasis (VL) incidence from data at different spatial scales.

However, here are a few comments and suggestions on the introduction.

3. Page 2, lines 48–52: the authors mention the MAUP. I think it would be ideal to explain further what the MAUP is for a non-technical audience and readers who may not be familiar with the concept.

Thank you for the suggestion. This point has now been expanded as follows:

[Line 50; Background] *It has been demonstrated previously that the choice of spatial scale/units of analysis can have an unintended influence on conclusions (known as the Modifiable Areal Unit Problem, or MAUP). In brief, this is the problem that a single set of data may present - in theory - infinitely many different patterns depending on the particular way it is partitioned and aggregated into discrete spatial units. As a result, different conclusions may be drawn from the same data if re-aggregated to a different set of areas.*

4. I think the writing in general is clear and concise, but there are a few instances where the phrasing could be improved for clarity and flow. E.g.

Line 47: "The vast majority of these villages are not affected by VL, in particular those south of the Ganges River, while others persistently observe cases and suffer outbreaks."

Line 65: "Interventions could be applied more efficiently if sporadically-affected villages were

covered within the range of a nearby persistently-affected village, rather than waiting for a response to be triggered within each independently”.

Thank you for the suggestions. The sentences have been rephrased as follows:

[Line 62] *In particular across the region south of the Ganges river, the vast majority of villages are not affected by VL. However, an important minority of villages suffer persistent outbreaks.*

[Line 70] *It could be more efficient to apply interventions within a certain range of a persistently-affected village, rather than waiting for a case to be detected in each individual village in order to trigger a response.*

Method:

5. On page 4, line 118, the authors stated their utilisation of population data from WorldPop. However, WorldPop offers various types of population data. Could the authors please specify which data product they utilised and from which year it was extracted? Clarification on this detail would enhance the comprehensiveness of the methodology and aid in reproducibility.

Apologies for the omission. This has detail has been added:

[Line 122; Materials and Methods] *Populations were estimated by first extracting and summing 100m pixel values from the WorldPop UN-adjusted population count raster for 2018 [23] for the set of village polygons.*

6. On page 4, line 126, it would be beneficial to clarify the year of the covariates utilised in the study. Additionally, the authors may want to provide justification for the selection of these covariates as input variables. How many covariates were considered, and was there a specific criterion for eliminating certain covariates?

This information has been added:

[Line 136; Pixel-level covariates] *Raster data for elevation (metres above sea level) and distance (in metres, as of 2015) to inland water bodies were obtained from WorldPop at a resolution of 100m for the region of Bihar state.*

Estimated travel time (in minutes) to the nearest urban centre as of 2015 was obtained from the Malaria Atlas Project (MAP) at a resolution of 1km.

The latter two were initially extracted on a monthly scale for the period of the case data (2018-2019) and subsequently aggregated to an annual mean and standard deviation.

These covariates were selected based on a literature review of environmental and climatic factors potentially associated with VL incidence. The characteristics suitable for sandflies and VL transmission that have been highlighted in previous work fell under the

broad categories of elevation, rurality versus urbanity and moisture in the environment (proximity to water, vegetation and temperature). This has been explained at the beginning of this section:

[Line 130] *Potentially predictive spatial covariates were identified by reviewing previous analyses of risk factors for VL incidence on a local population/community level, and characteristics of suitable sandfly habitats [24,25,26]. These broadly fell under the categories of elevation, rurality and moisture in the environment (proximity to water, vegetation and temperature). Publicly-available data sources were then identified to capture these characteristics at a pixel level with highest resolution.*

Furthermore, were there any concerns regarding multicollinearity among the covariates? Addressing these questions regarding covariate selection and potential issues would enhance the thoroughness and robustness of the study.

Multicollinearity is to be expected when considering closely linked environmental and climatic variables. However, if present, this would only affect the interpretation of individual associations rather than prediction. Moreover, the penalisation that is specified in the priors of the disaggregation models and the non-parametric nature of the random forest model mean that variables which are mostly represented by the information in other variables will not hold substantial weight in the model fit.

Correlation between variables and the interpretation individual associations is mentioned in the discussion:

[Line 380] *In this analysis, only annual variation in the vegetation index had a robust association with incidence. Such an association could be linked to differences in agricultural practices between higher and lower incidence regions; however, likely correlation between covariates means that individual effects should not be over-interpreted.*

7. On page 5, lines 140–144, the assessment of spatial autocorrelation appears unclear. It would be helpful to specify whether the spatial autocorrelation of VL incidence was evaluated at the block level or the village level. Please clarify

The assessment of spatial autocorrelation was conducted on the village level, both statewide and separately within each block. This has been clarified:

[Line 166; Descriptive analysis] *Preliminary analyses assessed the evidence for (global) spatial auto-correlation in incidence between neighbouring villages for the year 2018, by calculation of Moran's I statistic. This was calculated across all villages in the state and separately across villages within each block independently, to explore whether the strength of correlation between nearby villages could differ between blocks.*

8. On page 6, lines 146–153, there are different approaches to disaggregating observed incidence. Is there any justification for selecting the method they used in the disaggregation model? If so, clarify. Another thing is to mention the name of the package and what software was used for the disaggregation model

We aimed to evaluate one particular approach to disaggregation that is implemented in the R package *disaggregation*. This work was a collaboration with those who developed this package, to evaluate its utility with a real-world example. We acknowledge that there are other methodological approaches that could have been taken.

Thank you for flagging the omission of the package name. The package is now named alongside the citation of the article published in the *Journal of Statistical Software* (Nandi et al. 2023) to describe the methodology and implementation:

[Line 174] *Disaggregation regression combines observed block-level case counts with these finer-scale population and covariate data to predict the potential within-block distribution of incidence. For this analysis, we evaluate the implementation described in [13] and published in the R package disaggregation.*

9. Also, the covariates extracted for the modelling are measured in different units. Are these covariates scaled before they are used as input in the modelling? Unscaled covariates may have implications for the disaggregated result. I would suggest that the covariates used in the model need to be scaled or standardised into a z-score or some other method to allow for easy comparison of these covariates.

Yes, the covariates were all scaled for inclusion in the model. This is now stated:

[Line 146] *All covariate rasters were resampled to the lowest resolution (1km) and scaled to account for the inconsistent units of measurement.*

Validation

7. On page 6, lines 170-174, the description of the baseline prediction model for validation lacks clarity. It is unclear whether the model utilises the observed incidence rate at the block level and disaggregates cases across all villages, regardless of whether a village has previously recorded a case.

The definition of the baseline prediction has been expanded to improve clarity:

[Line 221] *First, the observed block-level incidence rate was defined as a baseline prediction for all villages within the block. This reflects the accuracy of assuming village-level incidence based on crude block-level surveillance, with no further information on heterogeneity between villages other than population size.*

If this is the case, then it is necessary to reconsider the baseline model for validation. If a village has not reported any cases, the model should be constrained accordingly and not make predictions in such a village. Redistributing case counts to villages with no previous incidence may not be appropriate because we are redistributing case counts to villages with no incidence in reality. If there are no recorded cases in a given village why are you making predictions at the pixel location in such villages? So, I think this model needs to be reconsidered.

As explained in the introduction [Line 85], our aim with this analysis was to investigate whether the pattern of cases between villages can be inferred from data collected at a coarser spatial scale. We attempt to do this by exploiting prior understanding of how incidence of this vector-borne disease is linked to the local environmental conditions that have been mapped on a fine spatial scale.

The fact of whether or not cases were observed in a particular village is *the exact thing we aim to infer*, using data from the block-level and the remotely-sensed characteristics of that village. It would therefore not be appropriate to constrain the predictions based on known incidence, as this would mean predicting the outcome from the outcome itself. If we applied this constraint, the resulting predictions would be artificially accurate as we have a priori told the models where the cases are.

11. On pages 175–177, I think the authors need to reconsider the spatial scale in the use of the random forest model in validating their predictions. The authors claim they fitted a random forest (RF) model at the village level and made predictions at the village level. And this was used to validate the model.

I think we have an inherent bias with the spatial unit of analysis. Ideally, if we want to compare how the RF model performs with our model, which is fitted at the block level, then the spatial unit of fitting the two models should be the same, i.e., the RF model should also use data observed at the block level and not at the village level as in the case of the RF. We want the spatial unit for fitting the model to be the same so that we are not introducing any bias in the predictions. So, the authors need to reconsider the RF spatial unit for fitting the model.

Our expectation was that a model fit directly to village-level incidence would offer the greatest accuracy for predicting village-level incidence based on the given covariate information, hence we wish to ascertain how close to this we get by using spatially-coarser data and a disaggregation approach.

The intention of fitting the RF model is therefore to assess whether it's possible, if we *did* routinely have access to data at the village level, to infer VL incidence for a particular village from other villages based on the given covariates. In other words, is the information contained in the given covariates sufficient to predict the VL incidence in a particular village? This is not meant to be another approach for disaggregation of the block-level observations.

Also, what package or software was used in fitting the RF model? A random forest model works best when you have more covariates to partition the feature space. I can't say how efficient the RF model will be with this small number of covariates. But I am guessing that could be a potential reason why the RF model metrics did not improve much relative to the other model. It would be worth considering other covariates in the RF model.

The package used to fit the random forest model has now been specified:

[Line 225] *Secondly, a random forest model [34,35] was fit directly to the village level data (using the R package randomforest) to serve as a "gold-standard"...*

We chose to use a RF for its robustness and flexibility, rather than its ability to navigate a large number of covariates (there is no inherent issue or loss of efficiency with training a RF on only a few covariates). Since our covariates are environmental and demographic characteristics they are likely interlinked in complex and non-linear ways; a non-parametric RF approach can help identify informative interactions and non-linear associations in this situation. Our aim is to *maximise* the information that can be gleaned from the given covariates about village-level incidence by allowing flexibility to identify the most informative associations (even if they are complex and non-linear).

It would not be appropriate to add further covariates to this model that are not present in the disaggregation model as this would make the predictions incomparable; they would then be based on different covariate information *in addition to* the different spatial scales we aim to compare.

12. Overall, I suggest that the authors revisit their validation approach. To effectively assess the efficiency of the disaggregation model in comparison with the RF model and other models, it is essential for all models to be fitted at the same spatial unit or scale. For instance, fitting all models at the block level and subsequently disaggregating them to pixel levels would ensure consistency and minimise bias in the comparisons.

As explained above, one of the key comparisons of interest is the predictive accuracy of models fit at different scales, to ascertain if a finer spatial scale of routine data collection is necessary to understand the sub-block-level distribution of disease burden. The objective of the analysis is therefore to fit models at different spatial scales and compare between these.

Furthermore, it is advisable to constrain the models so that predictions or disaggregation are only performed for villages with recorded VL cases rather than those without any incidence.

As explained above, I disagree with the recommendation to constrain the predictions – we are trying to predict what the distribution of burden is across villages *assuming we don't have data at the village level*. To constrain the model predictions to predict only in affected villages requires prior knowledge of where cases are, which is exactly what we are trying to predict.

Additionally, employing a variety of model metrics for evaluation beyond just RMSE and correlation would provide a more comprehensive assessment of model performance. This approach would enhance the robustness and reliability of the study's findings.

An additional error metric is now included in Figure 3.4, and all metrics have been calculated to exclude blocks with zero total incidence. The table has been removed as the values of each metric with and without zero blocks are more easily interpreted through the figure. The accompanying text has been updated with respect to the new figure.

[Line 315] *Out-of-bag predictions from the village-level random forest model attained a lower RMSE than the baseline, but were the weakest with respect to the other metrics considered. This pattern persists when blocks with zero observed cases are excluded from the calculations. The difference in ranking between RMSE and MAE suggests that the disaggregation models make some larger errors than the random forest model (which are penalised more strongly by RMSE), even if overall the predictions are closer to the observed values. As expected, the errors and correlation increase and decrease, respectively, with the exclusion of villages within zero-count blocks; as these form a substantial proportion of the total villages, accurately predicting zero cases here has a strongly influence on the summary measure.*

Results

13. On page 9, Data Cleaning, I think the data cleaning section should be shifted to the methodology section and not the result section.

This section has been moved.

12. On Page 11, lines 266–268, I think the authors need to specify in the methodology the different models they fitted with the disaggregation model. No mention is made of the fitting of two separate disaggregation models in the method, but the authors in the result section are reporting two disaggregation methods, which is confusing. They need to provide clarity in the method section.

Apologies for the oversight. This has been added:

[Line 209; Disaggregation model structure] *Two variants of the disaggregation model were fitted - both including and excluding the spatial random field. Comparing the estimated covariate effects in the presence and absence of the spatial field indicates the extent to which the field absorbs the spatial variation represented by the covariates.*

13. Page 11, table 3.2: I think the validation of the model should be reconsidered based on the comment I provided in the methodology section. Also, aside from the two metrics of correlation

and RMSE, different metrics can also be calculated to give a broader perspective of how the model is performing.

As stated above in response to point (12), an additional error metric has been presented.

Discussion

14. I think the author's discussion is well written overall. However, I think their discussion needs to be re-considered in line with key issues found within the results that relate to the methodology. I think if issues raised in the methodology are considered (absolutely at your discretion and that of the editor), then the discussion can be rewritten considering new results to communicate the new result that has been found.

This point is addressed within the responses to points (11) and (12) above.

Reviewer #3 (Remarks to the Author):

Summary:

The article evaluates the feasibility of using a statistical model to obtain accurate disease incidence rate estimates for individual villages based on aggregated incidence data at a coarser resolution (the block level). In particular, the authors study the incidence of visceral leishmaniasis in the Indian state of Bihar of 2018. The authors use a dataset of village level incidence rates which allows them to evaluate a disaggregation approach which only use the spatially aggregated block level data as input. The authors evaluate two geostatistical models that incorporate covariate information (derived from satellite imagery) and Gaussian process modeling, finding that the geostatistical models do not effectively downscale the aggregated data. Moreover, the authors show that even a random forest approach that utilizes the village-level data does not yield estimates that improve upon the baseline assumption of block-level homogeneity in incidence.

Overall impression:

The article is well-written, with a clear motivation and thoughtful discussion. The study's use of high-resolution spatial data to validate disaggregation approaches is highly valuable and relevant to researchers applying geostatistical modeling to estimate health-related rates, as such "gold-standard" datasets are rarely available. The presentation of the statistical and machine learning approaches used could be improved. In particular, additional justification of modeling choices and model validation would strengthen the article, as it is currently not obvious that the model used for disaggregation represents the best possible model. The discussion of results and limitations also needs editing to remove imprecise language, and the limitations related to the data (limited covariate data resolution, uncertainty related to the population estimates) should be emphasized.

Thank you for your summary and suggestions to improve the manuscript.

Specific comments and recommendation:

1. Line 154: The Gaussian random field is not clearly specified here. The authors should describe what covariance function is used, and how the parameters for the covariance function are estimated/specified.

Definition of the covariance function has been added to the text and greater detail on the estimation process has been added in the supplementary materials:

[Line 184] ...where X_{ij} are covariate values for pixel j in block i , GRF is a Gaussian random field with Matern covariance function defined across pixels s_{ij} , and u_i is a block-level uncorrelated random effect. This approach draws on and adapts the approach described in [33] for modelling spatial processes using R-INLA. Priors for each parameter and hyperparameter to be estimated are defined in the *Supplementary Materials*. Prior distributions for each parameter and hyperparameter to be estimated are defined in the *Supplementary Materials*.

[Supplementary materials]

Model fitting

The model structure defined in 2.4 requires a non-standard estimation procedure as the predictions to be obtained are at a different scale to the response data. In other words, the number of rows of covariate data differs from the number of rows of response data.

The disaggregation package defines the joint likelihood function and prior model in C++, which is then passed to Template Model Builder (TMB). TMB then uses a series of packages for automatic differentiation, linear algebra and computation of sparse matrices, to implement the automatic Laplace approximation [49] to obtain an approximation to the Bayesian posterior.

Priors

Fixed effects

$$\beta_0 \sim N(-4, 8)$$

$$\beta_k \sim N(0, 0.4)$$

Random effects - block-level

$$u_i \sim N(0, \sigma_u^2)$$

where σ_u is assigned a penalised complexity prior [50] such that

$$P[\sigma_u > 0.1] = 0.01$$

This shrinks the model towards the simplest case of no block-specific variation, except that which arises from the fixed covariate effects.

Random effects - Spatial field

The spatial random field is parameterised by the scale σ and range ρ of the Matern covariance function, which are again assigned penalised complexity priors as in [51],

$$P[\sigma > 2] = 0.01$$

$$P[\rho < 0.1] = 0.01$$

This shrinks the field towards the simplest 'flat' case, with zero variation and infinite range.

2. Line 158: The authors use a Poisson likelihood but do not justify this choice. Especially since there are many zeroes at the village level, would another likelihood or model be more appropriate? Is there any overdispersion observed? Some discussion of alternative choices (and subsequent justification of the model used) would improve the article.

An expanded explanation of likelihood choice has been added to this section:

[Line 191] A Poisson likelihood is a natural choice for a count outcome, and is mathematically convenient for the aggregation step within the disaggregation model structure. If we assume that the number of cases observed in each pixel follows a Poisson distribution, then it follows that the sum across pixels also follows a Poisson distribution.

t

It is often the case that greater variation is observed in the outcome of interest than can be accommodated with a Poisson distribution (overdispersion). The fitted IID component provides the flexibility to absorb residual extra-Poisson variation in block-level counts that is not explained by the given covariates, similar to the addition of a dispersion parameter within a Negative Binomial model. Conditional on the given covariates and random field, there was no evidence of residual extra-Poisson variation in the outcome.

3. Line 159-162: What are the parameters described by the posterior distribution here? It is true that the Laplace approximation relies on Gaussian assumptions for the posterior distribution, but it is not obvious that the posterior distribution for the $r_{\{ij\}}$ rate variables would be multivariate

Gaussian. This point could be further clarified. The inferential framework should also be discussed further. The mention of a posterior distribution suggests a Bayesian approach, but no prior distribution is mentioned and usually priors are not included when applying TMB.

For a more detailed, technical explanation of the inferential framework, we would refer the reviewer to the cited paper by *Nandi et al.* The explanation of the inferential framework has been expanded in the text and a more detailed specification given supplementary materials (as in response to point (1)). A further point on approximation of the posteriors has also been added to this section of the methods.

[Line 176] *For this analysis, we evaluate the Bayesian implementation described in [13] and published in the R package disaggregation. This approach draws on and adapts the approach described in [33] for modelling spatial processes using the integrated nested Laplace approximation (INLA).*

[Line 205] *Experiments in [13] and [23] have shown that the posteriors of the model's parameters and hyper parameters are well approximated by Gaussian distributions, using smaller datasets than that which is considered here.*

In response to the comment about priors in TMB, we would highlight that standard usage of TMB is within a *frequentist* framework. However, it is a simple (and widely used) extension to include the prior model in the model definition. The commissioner methods that would then calculate a maximum likelihood estimate and a normal approximation of the likelihood surface can instead be used to calculate a maximum a posteriori estimate and a normal approximation to the posterior.

4. Line 164: The aggregation step could be further explained. What is done with pixels that are partially overlapping village polygons?

A sentence has been added to explain this process:

[Line 213] *Pixel level estimates of incidence from the disaggregation model were weighted by the 1km population raster and aggregated according to the percentage coverage of each pixel overlapping each village shape, using the R package exactextractr. The resulting estimated case counts over each village polygon were then rescaled by the estimated village populations for comparison with observed incidence rates.*

5. Line 178: Typo; remove second "is."

Thank you for highlighting this. It has been corrected.

6. Line 191: Four alternatives are mentioned, but only three are provided.

Thank you for highlighting this. The error has been corrected.

7. Line 199: The claim that each observation is predicted 200 times is confusing. Are these observations held out of every tree? If a given observation is included in the training data for any of the trees, then it would be predicted fewer than 200 times.

Thank you for highlighting this error. It has been corrected:

[Line 250] *In this case, each observation is excluded (and hence predicted out-of-bag) an average of 74 times across the 200 fitted trees.*

8. Line 245: The authors note that villages cover an area of one to two square kilometers. If the covariate raster pixels are 1km by 1km, then perhaps the covariate data is simply not available at a high enough resolution to carry out disaggregation to the village level. Are there many villages that are smaller than a pixel? In such cases, how would the disaggregation procedure be justified? More discussion of how the resolution of the covariate data/prediction grid affects results is needed.

The reviewer raises an important point to consider. However, the approach would still work if some villages of a size around the same as a pixel – there would simply be a negligible difference between the pixel-level prediction and the prediction re-aggregated to the village shape (i.e. only one pixel would contribute to the final village-level prediction).

9. Line 252-253: “greater than two cases than expected” is unclear.

This sentence has been revised:

[Line 288] *In particular, observed incidence is more sparse and clustered, with a greater number of villages observing more than two cases than would have been expected from block level incidence rates.*

10. Figure 3.2: This figure is somewhat confusing. The total number of villages should be the same for both Baseline and Observed, but it looks like the total is much smaller for Baseline. Moreover, the choice of bins (of unequal size) makes the pattern look smoother than it may be in reality. I would suggest keeping all bins (0, 1, 2, 3, 4, 5, etc.) to improve interpretability.

This figure and the comparable figure 3.6 have been revised.

11. Line 286-288: Do all methods always predict zero cases for villages in a block with zero cases? It seems like it would be best to leave such blocks out of the analysis as no downscaling would be necessary if the analyst knew with certainty that there would be no cases in the block. As such, the authors should also provide error metrics that leave out these blocks.

Yes, the disaggregation models must predict zero incidence for villages within block that observed zero cases. This is the same for the baseline prediction applying an observed rate of zero at the block level to each constituent village. As suggested, the error metrics have been recalculated excluding blocks with zero observed cases (please see response to point (12) by reviewer 2 above).

12. Line 372: What does it mean that the “spatial random field will absorb this non-Poisson variation?” This should be made more precise and rigorous.

Further explanation of likelihood choice has been given in the methods section, as described above.

13. Line 384-385: Restricted spatial regression is typically used in association studies where estimating associations between covariates and response is a priority. In this case, only prediction is required. Would switching to restricted spatial regression affect the results here?

This point has been added:

[Line 418] *There may be scope for developing the current implementation of disaggregation regression to employ restricted spatial regression as suggested in [48], fitting the spatial random effects only within the residual space after adjustment for the specified fixed effects. As the goal for this analysis was prediction rather than inference of covariate associations, this was not investigated further.*

14. Line 422: “incidence appears largely stochastic” is too vague of a claim, given that VL has a clear mechanism of spread and infection. It may be true that it is difficult to predict incidence given available data, but this claim seems to suggest that the disease simply occurs randomly.

This statement has been rephrased to refer to reported diagnoses being stochastic, as opposed to underlying incidence:

[Line 459] *At this stage of near-elimination of VL in Bihar, reported diagnoses of VL appear largely stochastic. It is possible that relationships between the environment and transmission that naturally arise from the underlying biological mechanisms have been broken down by intensified control efforts, patterns becoming increasingly fragmented as incidence has fallen to very low levels. Cases continue to arise in within-village clusters, yet this does not appear to be informative of the detection of cases in neighbouring villages.*

12th September 2024

Communications Medicine

Dear Reviewers,

RE: “Inferring the distribution of Visceral Leishmaniasis incidence from data at different spatial scales”

Thank you very much for the further feedback on our submitted manuscript. Please find below our responses to your comments.

Kind Regards,

Emily Nightingale

Reviewer #3 (Remarks to the Author):

The authors have made a considerable effort to address issues raised in the first review and the exposition of the technical details and discussion of results are largely improved. In particular, the authors' efforts to explain their inferential framework and modeling choices are appreciated. Some issues remain; I provide some specific comments and suggestions below. The line numbers refer to the marked-up manuscript with changes tracked.

Specific comments and recommendations:

1. Line 55: The definition of the MAUP is confusing (and not correct—a finite dataset can only be partitioned and aggregated a finite number of ways). I recommend the authors precisely define the MAUP in concrete terms: for example, a model of village level data can yield different conclusions than a model of block level data.

This definition has been amended:

[Line 50] *It has been demonstrated previously that the choice of spatial scale/units of analysis can have an unintended influence on conclusions (known as the Modifiable Areal Unit Problem, or MAUP). In brief, this is the problem that a model of village-level data can yield different conclusions to a model of block-level data.*

2. Line 113: Typo: “Knowing the relative locations villages”

Corrected.

3. Line 203: The discussion of the Poisson likelihood is welcome. With regards to the iid random effect, the authors could mention that since the blocks are of different sizes, assuming the same variance for all blocks may be too simple. The authors note that “there was no evidence of residual extra-Poisson variation”—how was this investigated? Given this claim, it would be helpful to see parameter estimates for the various variance parameters discussed here.

The suggested point has been added to this section:

[Line 197] *It is often the case that greater variation is observed in the outcome of interest than can be accommodated with a Poisson distribution (overdispersion). One option to address this would be to use a negative binomial likelihood, with a fixed dispersion parameter to accommodate the additional variation which is estimated across all blocks. In this example, however, heterogeneity across blocks in terms of geographic area, population size and other characteristics would mean that assuming the same scale of extra-Poisson variation for all blocks is likely too simplistic.*

The residual variation was informally assessed by eye, to check that the magnitude was not excessive to what was expected. For simplicity, the comment about residual extra-Poisson variation has been removed.

4. Line 217: This comment is not particularly informative. It’s true that the estimated covariate effects change depending on whether the GRF is included, due to spatial confounding. It’s not clear what it means for your conclusions when you say the field “absorbs the spatial variation represented by the covariates.” The authors could more clearly explain the expected result from including/excluding the spatial random field.

This comment has been removed and replaced with the following:

[Line 214] *Two variants of the disaggregation model were fitted - both including and excluding the spatial random field. As the field offers greater flexibility in representing a spatial pattern, it was expected that this component would be more influential in the model if the selected covariates were not strongly informative. Substantial differences in the estimated fixed effects when the field is included or excluded could indicate that there are important features of the observed spatial pattern that are only represented by*

the random field, with the selected covariates appearing significant only as proxies in the absence of other information.

5. Line 238: typo: “Random forest is a non-parametric approach is commonly” **Corrected.**

6. Figure 3.2: As far as I can tell, there are still significantly more total observed villages (summing over the heights of the gray bars) than villages as predicted by the baseline prediction (summing over the heights of the blue bars). Why is this the case?

This is just due to the log scaling on the y-axis and the large number of villages predicted to have zero/one case. More villages are predicted to have zero/one cases than was observed, but the scaling makes the difference in the other direction at higher numbers of cases appear larger. A comment has been added to the figure caption to highlight this:

[Figure 3.2] *Observed village incidence for 2018 compared to that which would be expected assuming uniformity of incidence across each block (block incidence rate multiplied by village populations size, rounded to the nearest whole case). Note that the log scaling on the y-axis exaggerates the under-estimation of higher case counts compared to the over-estimation of villages with zero or one cases.*

7. Line 351 add space: “considered.This”
Corrected.

8. Figure 3.6: As with Figure 3.2, there are still significantly more total observed villages (summing over the heights of the gray bars) than villages as predicted by the various methods. Why is this? On another note, it may help to discuss why the observed density is shifted so far right compared with the prediction methods? It seems that these methods smooth incidence too much—so cases that in reality cluster in a small number of villages are redistributed too evenly across the villages in a block.

We agree with the reviewer’s interpretation. The explanation of this figure has been expanded to include the suggested point:

[Line 342] *The distribution of out-of-bag predictions more closely replicated the observed than the disaggregation-based predictions (Figure 3.6), but still under-predicted overall. Overall, all the methods considered here fail to account for the strong clustering of cases and instead redistribute cases too evenly across villages in a block, resulting in under-estimation of the possible number of cases within a village.*

9. Response to author's rebuttal about 1MB: It's true that 1MB is typically used in a frequentist framework, but hyperparameter priors may be included in the model files. However, base TMB code does not integrate over multiple hyperparameter values—rather, when a prior is included, inference typically follows an empirical Bayesian framework. Uncertainty from the hyperparameters is not propagated to the final posterior (and indeed can make a Gaussian approximation seem more appropriate than is really the case).

With respect to a Gaussian approximation seeming appropriate when it is in fact not, this is always going to be subjective and a trade-off in terms of computation. Here, the exact uncertainty around the estimates of the iid and field are not excessively important, by which we mean the major results of the analysis do not rely on them being perfectly calibrated. Instead, we are mostly focussed on the broad strokes of the models predictive ability.

We agree with the limitations that the reviewer raises and have emphasised these in the relevant sections:

[Line 183] *For this analysis, we evaluate the Bayesian implementation described in [13] and published in the R package disaggregation. This approach draws on and adapts the approach described in [33] for modelling spatial processes using the integrated nested Laplace approximation (INLA). Notably, under this approach the hyperparameters are handled similarly to an empirical Bayesian framework and therefore uncertainty in these parameters are not fully propagated to the posterior.*

[Line 210] *Experiments in [13] and [23] have shown that, while the uncertainty in the hyperparameters are not fully propagated, the posteriors of the model's parameters and hyper parameters are well approximated by Gaussian distributions, using smaller datasets than that which is considered here.*